# Optimizing Movement Performance with Altered Sensation: An Examination of Multisensory Inputs

**DOI:** 10.3390/brainsci13091302

**Published:** 2023-09-09

**Authors:** Niyousha Mortaza, Steven R. Passmore, Cheryl M. Glazebrook

**Affiliations:** 1Program of Applied Health Sciences, University of Manitoba, Winnipeg, MB R3T 2N2, Canada; mortazan@myumanitoba.ca; 2Faculty of Kinesiology and Recreation Management, University of Manitoba, Winnipeg, MB R3T 2N2, Canada; cheryl.glazebrook@umanitoba.ca

**Keywords:** paresthesia, reaching, goal-directed aiming, somatosensory, auditory feedback, kinematics, visuomotor

## Abstract

Two experiments were conducted to assess the impact of induced paresthesia on movement parameters of goal-directed aiming movements to determine how visual and auditory feedback may enhance performance when somatosensory feedback is disrupted. In both experiments, neurotypical adults performed the goal-directed aiming task in four conditions: (i) paresthesia—full vision; (ii) paresthesia—no vision; (iii) no paresthesia—full vision; (iv) no paresthesia—no vision. Targets appeared on a computer screen, vision was obscured using visual occlusion spectacles, and paresthesia was induced with a constant current stimulator. The first and last 20% of trials (early and late performance) were compared to assess adaptability to altered somatosensory input. Experiment 2 added an auditory tone that confirmed successful target acquisitions. When compared to early performance in the no-paresthesia and no-vision conditions, induced paresthesia and no vision led to significantly larger endpoint error toward the body midline in both early and late performance. This finding reveals the importance of proprioceptive input for movement accuracy in the absence of visual feedback. The kinematic results indicated that vision could not fully compensate for the disrupted proprioceptive input when participants experienced induced paresthesia. However, when auditory feedback confirmed successful aiming movements in Experiment 2, participants were able to improve their endpoint variability when experiencing induced paresthesia through changes in movement preparation.

## 1. Introduction

Human movements are planned and executed by perceiving and integrating visual, somatosensory, and auditory sensory inputs [1,2]. The two-component, multiple process model of limb control proposes that effective limb control for goal-directed aiming movements includes integration of expected and actual sensory inputs from multiple sensory inputs. These sensory inputs include expected sensory consequences that are based on sensory memories. The model is based on extensive research on the role of vision for controlling upper limb goal-directed movements, as vision is a dominant source of information for aiming movements [3,4,5].

The visual inputs from spatially accurate information to the central nervous system are a rich source of sensory information for movement control, including details of the limb, environment, and targets. A substantial body of literature has delved into the manipulation or elimination of each of these three components of visual input to elucidate their specific roles in movement planning and execution [6,7]. Studies that have focused on visual feedback related to the environmental cues and movement background have revealed that peripheral vision predominantly supplies online visual information about a limb relative to the environment, including the limb speed and direction as it approaches the target [7,8,9]. 

In contrast, visual feedback of the limb via central vision holds particular significance for regulating spatial positioning of the limb, especially as the limb approaches the target vicinity during aiming movements. Additionally, contingent on the time available for processing aiming movement tasks, as well as the number of practice trials, central vision can contribute to real-time (i.e., online) or offline control of movement direction [9]. 

The role of target vision has been studied independently and in conjunction with visual feedback of the limb. Removing visual feedback of the target leads to the movement unfolding based on the sensory memory of the target’s location [10,11,12,13,14]. This sensory representation can be used for online corrections if visual feedback of the limb is present [13]. Abahnini and Poteau (1999) demonstrated that even brief exposure to vision of the limb relative to the target significantly influences the planning of limb-target regulation later in the movement [8]. Based on the aforementioned literature and in conjunction with the multiple process model (see [2] for the multiple process model of limb control), vision of the limb serves as a reference point for movement planning and a crucial source of sensory input for online corrections.

Somatosensory information, and more specifically proprioception, has a complementary role in the control of human movement by providing information about the location of the limb in the space as well as the relative location of limbs to each other [15,16,17,18]. Thus, it is important to understand how these two major sources of information are integrated for voluntary limb control, including how humans adapt to changing sensory inputs. It is suggested that a combination of proprioception and visual inputs are the main sources of information for limb-target regulation. Thus, effective limb control includes integration of expected and actual sensory inputs from multiple sensory sources [2]. Knowledge about the individual and integrative role of different sensory input is crucial for rehabilitation programs when either of these sensory inputs are deficient. 

Previously, we used induced paresthesia to interrupt somatosensory inputs while also manipulating visual information by removing target vision upon movement initiation [19,20]. In this previous work [19], we exclusively manipulated target vision. Thus, participants could combine available limb vision with their sensory representation of the target location. The presence of limb vision can allow for limb-target regulation based on the memory of the target position. Furthermore, we explored the removal of target vision both with and without induced paresthesia to better understand the changes in motor control strategies during rapid and accurate movement execution while experiencing paresthesia. That study aimed to uncover how participants adapt their movement control strategies in the presence of induced paresthesia and vision of the limb. Our findings indicated that the absence of target vision coupled with paresthesia adversely affected motor performance accuracy and efficiency. When vision of the target was removed and paresthesia was induced, participants had significantly higher bias for errors towards the midline of their body early during practice when compared to the no-paresthesia condition, or when vision of the target was available with paresthesia. Together, this pattern of results indicates that vision of the target and intact proprioception both contribute to limb-target corrections. We also found that participants did adapt to the effects of induced paresthesia with increased practice by adjusting their movement strategy to rely more on pre-planned movements. The strategy of pre-planning their movements included reducing their initial movement impulses and the need for online corrections to counter augmented neuromuscular noise.

While the strategy of pre-planning movements when experiencing induced paresthesia was successful when vision of the limb was still available, it is unclear if this strategy will still be effective if vision of the limb is also removed. The current study serves as a continuation of our prior research and introduces novelty by eliminating vision of the moving limb and surrounding environment, thereby accentuating the significance of afferent somatosensory input from the limb for successful target aiming movements. The overall aim was to assess if and how participants could adapt to the altered somatosensory sensory input when vision target and the limb were removed at movement initiation. In particular, if participants would be able to pre-plan their movement with disrupted somatosensory, target, and limb visual inputs. Two experiments were conducted to explore how humans adapt to the altered sensory input of the limb. In Experiment 1, we examined the implications of eliminating visual input of the entire visual environment (including the moving limb and target) combined with the introduction of temporarily induced paresthesia. This experimental setup effectively emulated the conditions someone with paresthesia might encounter while moving in a dark environment. The primary aim of this manipulation was to discern whether visual input, specifically pertaining to the limb, plays a pivotal role in enhancing movement performance when participants are exposed to induced paresthesia. We hypothesized that removing visual input from the entire visual environment would lead to larger and more pronounced differences between conditions with and without visual feedback, as compared to our previous study [19]. This hypothesis stemmed from the understanding that removing limb-specific visual cues would prevent using limb-specific visual feedback to compensate for the induced paresthesia. However, we did expect proprioception information would be used to achieve some degree of online control when vision was removed [2,21]. Consequently, the presence of induced paresthesia was expected to disrupt these online control processes reliant on proprioception. Additionally, a comparison between early and late movement trials was conducted to probe participants’ adaptability to the somatosensory manipulation. Thus, a secondary objective of Experiment 1 was to investigate the changes in motor control strategies to these sensory manipulations.

Next, Experiment 2 was conducted to evaluate if augmented auditory feedback valuable to assist participants in updating their motor control strategies when experiencing disruptions in visual (target, limb, and environment) and somatosensory feedback. The auditory feedback provided participants with confirmation that their reaching had successfully acquired the target. Thus, the fundamental goal of Experiment 2 was to assess whether terminal auditory feedback could improve the development of accurate and efficient movements. We predicted participants would benefit from the addition of auditory feedback and would show reduced dependence on auditory versus visual feedback because auditory feedback is not the preferred modality [22,23].

## 2. Experiment 1

### 2.1. Materials and Methods

#### 2.1.1. Participants

Twelve healthy young adults (4 females, 8 males) with mean age of 22.9 ± 4.0 years participated in the current experiment. Participants had no neurological condition or orthopedic injury that would interfere with their performance of the task. All experimental procedures were approved by the local ethics board and all participants provided signed informed consent.

#### 2.1.2. Apparatus, Materials, Design, and Procedure

The methodology of the current experiment was identical to our previous study, with the exception of how visual feedback was manipulated [19]. Participants sat at a table in front of a 17” monitor positioned lengthwise and in line with the table-top (Figure 1). 

The task included upper limb goal-directed reaches to one of the four square shaped targets, all with an index of difficulty of 6 [24]. The tasks were completed in four sensory conditions: (1) full vision, paresthesia; (2) no vision, paresthesia; (3) full vision, no paresthesia; (4) no vision, no paresthesia. Participants completed the four conditions across two separate experimental sessions. That is, the sessions that included induced paresthesia (conditions 1 and 2) were spaced at least 24 h apart from the session without induced paresthesia (conditions 3 and 4). Moreover, a 5–10 min break was introduced between the two vision conditions within each session. Order of the conditions and the specific target allocated for each condition was randomized and counterbalanced across participants, with the caveat that both paresthesia conditions were completed during the same experimental session. 

Each trial began with participants positioning their index finger on the designated home position, marked by a microswitch positioned at the lower part of the screen (Figure 1). The experimenter triggered each trial after ensuring participants were prepared with their finger in place. Following this, a fixation cross appeared on the screen for an unpredictable foreperiod lasting between 800 to 1400 milliseconds. Subsequently, the appearance of the target indicated to participants that they should move their finger to touch the target’s location on the screen. In the no-vision conditions, vision of the target, limb, and the environment was obscured upon movement onset, as soon as the microswitch was released, using visual occlusion spectacles (Translucent Technologies, Toronto, ON, Canada). At the end of each trial, feedback was provided to participants, including their movement time and if they hit or missed the target. For each condition, participants accomplished 100 distinct reaching movements towards the designated target on the touchscreen, resulting in a total of 200 across two conditions trials per session. This accumulates to a combined total of 400 trials across the two sessions and four conditions. A Digitimer DS7AH constant current stimulator (IBIS 169 Instrumentation Canada, Inc., Ottawa, ON, Canada) was used transcutaneously to generate constant stimulation to induce paresthesia in the median nerve. The protocol used for inducing paresthesia has been reported previously [19,20]. To confirm that sensory perturbation was achieved, monofilament pressure sensitivity test results were compared at baseline and after induced paresthesia using the Touch-Test^®^ Sensory Evaluation kit (North Coast Medical, Inc., Morgan Hill, CA, USA; Table 1). Data were collected on two separate days. Movements were recorded using a 3D motion capture system at 300 Hz (Optotrak 3D Investigator, Northern Digital Inc., Waterloo, ON, Canada) by tracking an infrared emitting diode (IRED) attached to the participants’ index fingers. Data collection onset was synchronized with the computer task and lasted for 2 s using custom software designed in E-Prime (version 2.0, Psychology Software Tools, Inc., Sharpsburg, PA, USA). The movement data were reduced and analyzed using a customized MATLAB program (version 8.1 (R2013a), the MathWorks, Inc., Natick, MA, USA). Movement kinematics were smoothed with a dual-pass Butterworth filter with a cutoff frequency of 15 Hz. Movement onset was marked as the first frame that the IRED moved faster than 30 mm/s for at least 30 ms, and movement offset was identified as the first frame that the IRED’s velocity fell below 30 mm/s for 30 ms [19].

The main outcome measures included movement time (MT), reaction time (RT), peak velocity (PV), time to peak velocity (ttPV), time after peak velocity (taPV), and ttPV normalized by MT (ttPV/MT). Accuracy variables were calculated in the anterior-posterior (primary) and medio-lateral (secondary) axes including constant error (CE) and variable error (VE). Both the anterior-posterior and medio-lateral axes signified a positive direction when movements were away from the body. Undershoot errors for the anterior-posterior axis were designated as movements shorter than the target amplitude, which positioned them closer to the body. In a similar fashion, undershoot errors for the medio-lateral axis were identified as movement amplitudes shorter than the target location, resulting in the actual movement endpoint being closer to the body midline. Moreover, variability of movement trajectories in the primary axis was calculated. That is, spatial variability at 20%, 40%, 60%, and 80% of the movement time was analyzed (see [19] for details of the outcome measures calculations). In order to investigate the effect of practice on performance adaptability only the data from the first (early performance) and last (late performance) 20 trials were analyzed for all of the outcome measures.

#### 2.1.3. Statistical Analyses

Data analysis included a repeated measures ANOVA with the following designs: 2 paresthesia conditions (paresthesia, no paresthesia) by 2 vision (vision, no vision) conditions by 2 practice (early, late performance). The factor of percent of movement time (20%, 40%, 60%, 80%, 100%) was added to the ANOVA in order to assess spatial variability throughout the movement trajectories. Significant interactions were further analyzed using Tukey’s HSD. Wilcoxon signed-rank test was used to compare the results of monofilament tests before and after induced paresthesia. An effect size of 0.33 was intended, and the significance level was set at *p* < 0.05 for all analyses. Statistical analyses were performed using a combination of Microsoft Excel (version 16, Microsoft, Redmond, WA, USA) for initial data organization and preprocessing, followed by more advanced analyses conducted with the Statistical Package for the Social Sciences (version 28, IBM^®^ SPSS^®^ Statistics, SPSS Inc., Armonk, NY, USA).

### 2.2. Results

Monofilament test results were missing for one participant due to recording errors, so data for 11 participants were analyzed for baseline versus post-stimulation comparison using the Wilcoxon signed-rank test. The results showed that participants sensed thicker filaments after the nerve stimulation was applied compared to their baseline (Z = −3.21, *p* = 0.001, Table 1).

#### 2.2.1. Temporal Measurements

No significant interactions or main effects was found for RT (Figure 2a). For the outcome of MT, a significant interaction was found for vision and paresthesia, F (1,11) = 8.85, *p* = 0.013, ηp^2^ = 0.45. Tukey’s HSD test showed that only when there was no paresthesia did participants have longer MT with vision versus without vision. Additionally, when comparing the no-vision conditions with and without paresthesia, participants demonstrated significantly longer movement times in the presence of paresthesia (Figure 2b). 

For peak velocity (PV), there was also a significant interaction of practice and paresthesia, F (1,11) = 4.93, *p* = 0.048, ηp^2^ = 0.31. Post hoc analysis showed that without paresthesia, there was a significantly higher PV at the later practice trials versus early trials (Figure 2c). 

Statistical analysis for ttPV revealed a significant main effect for vision; ttPV was significantly longer when vision was removed, F (1,11) = 5.00, *p* = 0.047, ηp^2^ = 0.31. A significant two-way interaction for vision and practice was also found, F (1,11) = 9.97, *p* = 0.009, ηp^2^ = 0.47. In addition, a significant three-way interaction for vision, practice, and paresthesia was found, F (1,11) = 7.20, *p* = 0.021, ηp^2^ = 0.40, with Tukey’s HSD test showing significantly shorter ttPV without any sensory manipulation (with vision and no paresthesia) at late performance trials when compared to early performance with the same sensory condition, and with late performance trials of both no-vision conditions with and without paresthesia (Figure 2d). 

When ttPV outcome was normalized with MT, statistical analysis showed a main effect of vision; that is, participants spent a larger percentage of their movement time before PV when vision was removed compared to the full-vision condition, F (1,11) = 23.82, *p* = 0.000, ηp^2^ = 0.68 (Figure 2f). Findings of taPV showed a significant interaction of vision and paresthesia. Post hoc analysis showed that only in the no-paresthesia conditions, when vision was available, did participants spend more time after PV, while in the induced paresthesia conditions, manipulating visual input did not affect taPV (Figure 2e).

#### 2.2.2. Spatial Measurements 

The ANOVA showed no significant main effects or interaction for the outcome of CE in the primary axis of movement (Figure 3a). The results of the statistical analysis for VE showed a significant main effect for vision, F (1,11) = 72.35, *p* = 0.000, ηp^2^ = 0.87 (Figure 3b), indicating that participants had significantly higher VE when vision was removed. 

Results of CE in the secondary axis showed significant main effect for vision, F (1,11) = 80.06, *p* = 0.000, ηp^2^ = 0.88, indicating larger undershoot errors when vision was removed (Figure 3c). There was also a significant two-way interaction of paresthesia and practice, F (1,11) = 8.45, *p* = 0.014, ηp^2^ = 0.43, as well as a three-way interaction of paresthesia, practice, and vision, F (1.72,18.89) = 116.77, *p* = 0.000, ηp^2^ = 0.91. The three-way interaction showed that CE in the secondary axis for the no-vision and no-paresthesia condition was significantly smaller than all other no vision conditions, including late performance blocks. Results of VE in the secondary axis was similar to the primary axis; that is, there was a significant main effect for vision indicating more variability when vision was removed (Figure 3d).

#### 2.2.3. Movement Trajectories

Primary axis movement trajectory: There was significant main effect for factors of vision F (1,11) = 11.93, *p* = 0.005, ηp^2^ = 0.52 (Figure 4a); practice, F (1,11) = 9.96, *p* = 0.009, ηp^2^ = 0.47 (Figure 4b); and percent of movement time, F (1.7,18.9) = 116.77, *p* = 0.000, ηp^2^ = 0.91 (Figure 4a,b). Trajectories were more variable when vision was removed compared to full-vision conditions and at early performance compared to the late trials. Post hoc analysis for the significant main effect of percent of movement showed that variability was significantly higher at 40% compared to 20% and 80% of movement practice.

There was a significant two-way interaction of vision and percent of movement, F (2.3,25.2) = 33.48, *p* = 0.000, ηp^2^ = 0.75 (Figure 4a). Post hoc analysis showed that when vision was not available at 60 and 80% of MT, trajectory variability was significantly higher than the conditions with full vision. Another significant interaction was practice and percent of movement time, F (2.1,23.2) = 63.88, *p* = 0.038, ηp^2^ = 0.25 (Figure 4b). Post hoc analysis showed that at 40% and 60% of MT trajectory variability decreased significantly at late performance when compared to early performance.

There was a significant three-way interaction of vision, paresthesia, and percent MT, F (2.1,23.2) = 7.00, *p* = 0.004, ηp^2^ = 0.39 (Figure 4a). Post hoc analysis showed that in the no-vison and no-paresthesia condition, at 60% of MT, there was significantly more variability than the two vision conditions (VP and VNP). This was not the case for the no-vision with paresthesia condition at 60% into the movement, although there was a trend for higher variability with paresthesia and no vision compared to the vision conditions. Comparison of the trajectory variability at 40% versus 60% of movement in both no-vision conditions showed significantly higher variability at 40%.

### 2.3. Discussion

The results of the current experiment showed that removing vision of the limb and environment resulted in significant changes in motor control and performance characteristics of a goal-directed aiming task compared to the full-vision condition. Also, as predicted, these changes were greater when the no-vision condition was combined with interrupted proprioception using induced paresthesia. 

#### 2.3.1. Effect of Sensory manipulation on Endpoint Accuracy

Removing vision of the environment increased endpoint variability in both the primary and secondary axes, regardless of the paresthesia condition. There was no significant movement bias (CE) in the primary axis; however, descriptive results showed that when vision was removed, the average CE indicated undershot for the no-paresthesia condition (mean = −3 mm) and an overshoot for the paresthesia condition (mean = 4 mm; Figure 3a). In the condition with the most sensory disruption (i.e., no vision, with paresthesia) there was a significant movement bias towards the midline throughout early and late performance when compared to the early performance of the no-paresthesia condition. This finding could signify the role of proprioception in the online correction and limb-target regulation which led to less movement bias in the no-vision, no-paresthesia condition (i.e., with intact proprioceptive inputs). Late performance in the no-vision and no-paresthesia condition also had a significant bias towards the body midline when compared to the early performance. One explanation for these findings could be the bias of sense of limb position towards the attended side of the body in the absence of visual input of the limb. That is, several studies [25,26,27] have reported that in the absence of visual input of the right or left hand, individuals tend to overestimate their limb location further towards right or left, respectively. In the current study participants seemed to overcorrect this bias by moving the upper limb more towards the midline, which led to undershoot errors specifically in the condition with paresthesia. Despite the availability of intact proprioceptive feedback, without vision during a block of 100 aiming movements, the amount of exposure to visual information was reduced such that by the final 20 trials of the overall block of trials, a similar overcorrection phenomenon may have occurred. The above findings are consistent with the results of our previous study [19] in the condition without vision of the target and no paresthesia. However, there were some differences with this previous study in that with vision of the limb and environment the movement bias (CE) in the secondary axis for late versus early performance was not statistically significant. 

The tendency of the participants to undershoot the target in the no-vision and paresthesia condition in the secondary axis (Figure 3c) could also be caused by the stimulation of the median nerve that was used to temporarily induce paresthesia. Isolated stimulation of the median nerve may have caused illusory sensation, leading to a biased sense of limb position towards elbow and wrist extension and the tendency for overcorrection towards the midline (elbow and/or wrist) flexion when vision feedback was not available to correct. This explanation is consistent with the finding of Rangwani et al. [28], where they used transcutaneous electrical muscle stimulation over the synergist flexor muscles of the elbow. They found that all participants experienced proprioceptive illusory sensation towards elbow extension according to the results of an arm-matching test. This illusory sense of limb position towards the movement of the antagonist muscle groups of the stimulation side is also seen in tendon muscle vibration that targets the muscle spindles [29,30]. In summary, induced paresthesia exacerbated the movement bias in the absence of vision, possibly by illusory sense of position towards wrist/elbow extension, when consistency (VE) was primarily impacted by vision only.

#### 2.3.2. Effect of Sensory Manipulation on Movement Strategy

The MT results showed that the lack of vision only affected MT in the no-paresthesia condition. Specifically, participants spent less time executing their movements when vision was removed; however, this was not the case for the paresthesia conditions. In the no-paresthesia and full-vision condition, participants had the maximum amount of sensory information available for online correction, which led to a movement strategy of using the available feedback, which led to longer MTs. The longer MT for the full-vision and no-paresthesia condition was also accompanied by significantly smaller CE in the secondary axis and well as smaller VE in both axes. Together, these results indicate successful online corrections of the aiming movement. 

The above results are consistent with the MT findings from previous research [31] where participants spent a significantly longer time after PV when vision was available. Time to PV results also showed shorter ttPV for full-vision conditions and the least amount of ttPV spent in the late performance of the full-vision and no-paresthesia condition compared to late performance of the no-vison conditions, regardless of presence of paresthesia. Also, consistent with other temporal variable findings, normalized ttPV showed a smaller percentage of movement time was spent before PV when vision was available, compared to no-vision conditions. 

In summary, the results of the temporal variables indicate that when vision was not available, participants spent most of their MT time on the distance covering portion of the movement and performed fewer online corrections. This movement strategy makes sense as they did not have visual feedback available for limb-target regulations [2,32]. The notable difference that paresthesia made on movement execution was on the time spent after peak velocity. That is, in full-vision conditions, when paresthesia was not applied, significantly more time was spent on the online correction portion of the movement. In contrast, when paresthesia was applied, participants did not engage in the same amount of time in the online corrective phase [2,31]. This finding indicates that when proprioception was compromised (by induced paresthesia), participants had less sensory input to process as they were focused on visual feedback alone for online correction, which led to a shorter time spent on limb-target regulation. 

As expected, movement trajectories were most variable at 40% of movement time as well as at movement end when vision was not available. In the no-vision and no-paresthesia condition at 60% of movement, there was significantly more spatial variability than the two vision conditions (vision with paresthesia and vision without paresthesia), while both vision conditions were significantly more variable at 40% of movement time compared to 60% of movement time (Figure 4a). Forty percent of movement time corresponds approximately to the time that PV was achieved (PV at ≈38% of movement time for vision conditions) and is expected to be close to the end of the initial impulse phase of the movement [2,32,33,34]. Together, the above findings suggest when vision was available, participants used a shorter and more forceful initial impulse phase such that the limb reached closer to the target location, leaving more time for online corrections. This strategy led to more variability earlier during the movement in the vision condition (at 40% of movement time) compared to the no-vision and no-paresthesia condition (at 60% of movement time). This finding is also in agreement with the taPV results for the condition with no vision and no paresthesia. That is, this condition had the shortest taPV (i.e., less time spent on online corrections). The vision conditions stayed significantly less variable than no-vision conditions for the rest of the movement (at 80% of movement time). 

#### 2.3.3. Sensory Manipulation and Practice

A secondary objective of the current study was to assess the adaptability of the motor control processes to the changes of sensory inputs; early and late 20% of trials were also analyzed separately. The only significant effect of the factor of practice on the accuracy outcomes was in CE in the no-vision and no-paresthesia condition. CE in the secondary axis was significantly greater in the later performance trials compared to the early trials. As discussed earlier, this finding can be explained by an overcorrection in the movement as a result of the biased sense of limb position in the prolonged absence of vision of the hand [25,26,27].

An effect of practice was seen with a significant decrease in trajectory variability with more trials. This difference was the most pronounced at 40% of movement time (Figure 4b), which can be explained by more consistent and refined force generation profiles with repeated movements towards the same target [35]. Specifically, 40% of movement time corresponds to the time at which peak velocity is typically reached [36]. However, the difference in available sensory inputs did not have a significant effect on changes in spatial variability with practice.

Results of ttPV for early and late performance showed that when vision was available, participants performed the movements with shorter ttPV after practice when compared to their early performance. This change of movement strategy when vision was available was most noticeable when paresthesia was not applied. From the findings of the effects of practice on the length of ttPV, we infer that a change of movement control strategy from a more pre-planned movement to using more online control occurred when participants had all intrinsically available sensory information. In contrast, participants did not update their movement control strategy when somatosensory input was altered, which indicates the critical role of proprioception in online corrections even when full visual input was available. 

In summary, visual feedback about the moving limb had a significant effect on endpoint accuracy and variability; however, paresthesia or lack of intact proprioceptive input contributed to a pre-planned movement strategy for goal-directed reaching, even when full visual feedback was available. Therefore, at least within the single testing session and the current task parameters, participants did not adapt their movement strategies to the distorted proprioceptive input.

## 3. Experiment 2

When paresthesia was induced in Experiment 1, participants were unable to update their movement strategies to account for the altered proprioceptive input. Disruptions in somatosensory input due to injury and disease are a common experience. Experiment 1 provided additional evidence that these changes impact the performance of the types of goal-directed aiming movements that are used daily when interacting with touchscreen technology. Therefore, understanding how to facilitate performance when somatosensory input is disrupted has both theoretical and practical implications. A growing literature in both neurotypical and neurodiverse populations have provided evidence that augmented auditory feedback can facilitate movement performance [37,38,39]. When provided at or near target acquisition, augmented auditory likely works by providing confirmation to participants that they successfully acquired the target. Thus, the purpose of Experiment 2 was to assess if it is possible to supplement for the disrupted somatosensory feedback through providing this type of augmented auditory feedback at target acquisition. 

We predicted that the addition of an auditory tone would facilitate improvements in movement efficiency when visual feedback was not available. In other words, the benefit of the auditory feedback would be greater when both visual and somatosensory feedback have been disrupted. However, when two sources of accurate feedback are available (i.e., vision and audition) then movements will be performed more accurately compared to when only one source of accurate feedback is available. 

### 3.1. Methods

#### 3.1.1. Participants

Fourteen healthy young adults (5 females, 9 males) with mean age of 22.7 ± 2.9 years participated in Experiment 2. Consistent with Experiment 1, participants had no neurological condition or orthopedic injury that would interfere with their performance of the task. All experimental procedures were consistent with the Declaration of Helsinki and were approved by the local ethics board. All participants provided signed informed consent prior to their participation in the experiment and received a base honorarium for their time.

#### 3.1.2. Apparatus, Materials, Design, Procedure, and Analysis

The overall experimental set-up and design of Experiment 2 was identical to Experiment 1, with one difference: an auditory tone was introduced, sounding only when participants successfully landed on the target. The beep was presented immediately at target acquisition (i.e., when participants touched the target). This auditory feedback complemented the information about movement time and whether they hit or missed the target. The latter two types of feedback were the only ones provided in Experiment 1. Consistent with Experiment 1, vision and induced paresthesia were blocked and counterbalanced across participants. Thus, the experimental design remained the same, with the addition of auditory feedback for successful trials in all four conditions. The individual trial sequence also remained the same, with one exception. After participants completed each movement, an auditory tone was presented via standard computer speakers when that specific movement was accurate. The data analysis also followed the procedures outlined in Experiment 1. Please see Experiment 1 for details of the visual and somatosensory conditions.

### 3.2. Results

#### 3.2.1. Monofilament Test

As in Experiment 1, baseline monofilament test results were analyzed and compared with post-stimulation using the Wilcoxon signed-rank test. The findings revealed that participants sensed thicker filaments after the application of stimulation, in contrast to their baseline measurements (Z = −3.329, *p* = 0.001, Table 2).

#### 3.2.2. Temporal Measurements

No significant main effects or interactions were found for the factors of vision, practice, or paresthesia for MT (Figure 5a). For the outcome of RT, significant main effects were found for vision (F (1,13) = 12.936; *p* = 0.003, ηp^2^ = 0.499) and practice (F (1,13) = 16.283; *p* = 0.001, ηp^2^ = 0.556, Figure 5b). Reaction times were found to be significantly longer in the no-vision condition compared to the vision condition, as well as during early performance compared to late performance.

Regarding peak velocity (PV), a significant interaction between practice and vision was observed, F (1,13) = 7.307, *p* = 0.018, ηp^2^ = 0.36. Tukey’s HSD analysis revealed that while PV was significantly lower with full vision compared with no vision in early performance, with more practice in the late performance trials, PV with full vision increased and became closer to the PV value in the no vision in late performance (Figure 5c). Statistical analysis for time to peak velocity (ttPV) revealed a significant main effect for vision: ttPV was significantly longer when vision was removed, F (1,13) = 6.234; *p* = 0.027, ηp^2^ = 0.324. Also, there was significant interact ion of factors of vision and practice (F (1,13) = 6.388; *p* = 0.025, ηp^2^ = 0.329). Tukey’s HSD test showed that only in the late performance with full vision did participants have significantly shorter ttPV compared to when vision was removed; however, the ttPV in these two vision conditions was not significantly different in the early performance (Figure 5d). 

No significant main effects or interactions were found for the factors of vision, practice, or paresthesia for time after peak velocity (taPV; Figure 5e). When ttPV outcome was normalized with MT, statistical analysis showed a main effect of vision, F (1,13) = 6.364; *p* = 0.025, ηp^2^ = 0.329; that is, participants spent a larger percentage of their movement time before PV when vision was removed compared to a full-vision condition (Figure 5f).

#### 3.2.3. Spatial Measurements

The ANOVA for the outcome of CE in the primary axis of movement showed a significant main effect for practice, F (1,13) = 4.713, *p* = 0.049, ηp2 = 0.266, as well as a significant interaction for the factors of vision and practice, F (1,13) = 5.715, *p* = 0.033, ηp^2^ = 0.305 (Figure 6a). Tukey’s HSD showed significantly larger overshoots for the no-vision condition when compared with the vision condition in the late performance. However, the CE in the primary axis in these two conditions was not significantly different in the early performance. Also, according to Tukey’s HSD, comparisons of late versus early performance in the no-vision conditions found a significant increase in CE late in performance. Results of the statistical analysis for VE showed significant main effect for vision, F (1,13) = 133.209, *p* = 0.000, ηp^2^ = 0.911 (Figure 6b) and interaction of vision and paresthesia, F (1,13) = 5.308, *p* = 0.038, ηp^2^ = 0.290, as well as an interaction between paresthesia and practice (F (1,13) = 6.841; *p* = 0.021, ηp^2^ = 0.345). Tukey’s HSD for the interaction of vision and paresthesia showed significantly larger VE for the no-vision condition when compared with the full-vision condition in both with and without paresthesia conditions. Additionally, comparisons of with and without paresthesia trials when vision was blocked showed significant increase in VE when paresthesia was present. Post hoc testing using Tukey’s HSD for interaction of paresthesia and practice showed that participants had significantly larger VE in the presence of paresthesia when compared with the no-paresthesia condition only in the early performance; however, the VE in the primary axis in these two conditions was not significantly different in the late performance (Figure 6b). Additionally, comparison of VE in the early and late performance of the paresthesia condition showed significantly higher VE in the early performance.

Consistent with Experiment 1, results of CE in the secondary axis showed significant main effect for vision, F (1,13) = 18.62, *p* = 0.001, ηp^2^ = 0.589, indicating larger undershoot errors when vision was removed (Figure 6c). Results of VE in the secondary axis were similar to the primary axis; that is, there was a significant main effect for vision indicating more variability when vision was removed (F (1,13) = 232.338; *p* = 0.000, ηp^2^ = 0.947; Figure 6d). Additionally, a main effect for practice was observed (F (1,13) = 6.711; *p* = 0.022, ηp^2^ = 0.340; Figure 6d), indicating that participants exhibited smaller VE in the secondary axis with more practice.

#### 3.2.4. Movement Trajectories

The repeated measures ANOVA for the spatial variability in the primary movement axis showed significant main effects for percentage of the movement, F (1.372,17.842) = 81.553, *p* = 0.000, ηp^2^ = 0.863, and vision F (1,13) = 6.840, *p* = 0.021, ηp^2^ = 0.345 (Figure 7a). Also, there was a significant interaction between factors of percentage of movement and vision, F (1.824,23.712) = 16.316, *p* = 0.000, ηp^2^ = 0.557 (Figure 7a), and another interaction between the paresthesia and practice, F (1,13) = 13.578, *p* = 0.003, ηp^2^ = 0.511 (Figure 7b). Post hoc analysis for the interaction of percentage of movement and vision showed that at 80% of the movement time, no-vision conditions had significantly higher variability compared with the full-vision condition. Post hoc analysis for the interaction of practice and paresthesia showed that, in early performance, the presence of paresthesia significantly increased the spatial variability (Figure 7b). Also, when paresthesia was present, trajectories were more variable in early performance when compared with the late performance. 

### 3.3. Discussion

#### 3.3.1. Effect of Sensory Manipulation on Movement Strategy

Participants in Experiment 2 adapted their RT, PV, and ttPV to accommodate the changing visual and somatosensory inputs. RT, PV, and ttPV are all measures that are associated with adjusting movement preparation [2,31,40]. Specifically, with practice, participants increased their PV from early to late performance when vision was available. This increase in PV likely reflects a more confident movement plan that may also result in more efficient movements. That is, in early performance, participants reduced their PV to achieve more accuracy. As they practiced more, they improved their movement planning, as shown by shorter RTs combined with higher PV with vision and late performance [41]. Moreover, consistent with Experiment 1, ttPV was significantly shorter than the ttPV in the no-vision condition. This pattern indicates that only when vision was available did more practice lead to more forceful movement initiation, which led to a higher PV and shorter ttPV. When vision was obstructed, participants reduced their PV to mitigate the spatial variability of their movements and improve the endpoint accuracy of their limb movements [42]. More forceful movement initiation only when vision was available would be expected as participants could use visual feedback during movement execution to adjust for any increase in the variability of the initial movement impulse [2,41]. 

#### 3.3.2. Effect of Sensory Manipulation on Endpoint Accuracy

The analysis of constant error in the primary axis showed an expected effect for vision, whereby target aiming was more accurate when vision was available. The improved accuracy was also achieved within the same movement time as trials without vision. When vision was removed at movement initiation, there was a clear effect on endpoint accuracy. However, somewhat unexpectedly, participants displayed larger overshoot errors in the primary axis with more practice when compared with their early performance. Overshoot errors are not typically reported in the literature; with increased practice without vision, one would typically expect participants to undershoot the target. It is possible that the relatively high index of difficulty combined with the auditory feedback, and the task instructions designed to motivate participants to challenge themselves to move as quickly and accurately as possible, combined to encourage participants to take on a more forceful movement strategy.

Consistent with the above explanation, the significant interaction for vision and paresthesia for VE in the primary axis showed that lack of visual input led to even larger VE in the primary axis when paresthesia was present. In addition, the interaction of practice and paresthesia in the VE for the primary axis showed that when compared with the no-paresthesia condition, induced paresthesia led to significantly larger VE early in performance; however, with practice, participants improved their performance to the VE of the no-paresthesia condition. The interaction of practice and paresthesia for the movement trajectories also showed a similar result. That is, when paresthesia was induced, then more practice led to significantly less spatial variability. However, this same pattern was not seen for the no-paresthesia condition. The lack of improvement with practice for the no-paresthesia group likely occurred because participants were accustomed to integrating proprioceptive input from the limb and therefore did not need to adjust their movement strategies in the no-paresthesia condition. Finally, the interaction of percentage of movement time and vision indicates that visual feedback was being used to improve endpoint accuracy in the vision conditions, as at 80 precent of movement time there was reduced spatial variability in the vision condition. This reduction in spatial variability late in the movement is reported to represent the implementation of online correction based on the available visual feedback. 

#### 3.3.3. Sensory Manipulation and Practice

With more practice, participants in Experiment 2 showed significantly higher PV only when vision was available (Figure 5c). Also, when compared with no-vision conditions, ttPV became significantly shorter during late performance with vision (Figure 5d). This pattern suggests a shift in movement control strategy and planning. That is, moving from primarily pre-planned movements to a greater reliance on online control mechanisms when participants had access to their intrinsic sensory inputs. In contrast, participants did not alter their movement control strategy in response to changes in somatosensory input (induced paresthesia). These findings are consistent with the outcomes of Experiment 1. In Experiment 2, there was a notable positive effect of practice as observed in the results of VE in the primary axis in the no-vision condition and in the presence of paresthesia (Figure 6b); participants improved their accuracy by reducing their VE with more practice. This improvement in the endpoint variability could be attributed to the augmented auditory feedback in Experiment 2 (please see the general discussion for the direct comparison of the endpoint variability for the participants of the two experiments). Another effect of practice was seen in larger CE with more practice where in conditions without vision participants had more overshoot errors (Figure 6a). Auditory stimuli have been reported to have an activating effect [43] on movement performance, and therefore it is possible that the lack of visual feedback combined with auditory feedback led to more forceful movement impulses that resulted in target overshoots [44].

## 4. General Discussion

### 4.1. Adaptability and Practice

In both experiments, participants benefited most from practice when their intrinsic feedback was unaltered. That is, for both experiments, participants improved their movement strategy throughout practice by using a shorter ttPV and higher PV in the no-paresthesia and full-vision condition specifically. Thus, changes to movement strategies with practice that helped participants achieve movements that were both fast and accurate occurred primarily with unaltered intrinsic sensory inputs. That said, we did find evidence that participants adjusted their movement strategies when either or both visual and somatosensory feedback were disrupted.

### 4.2. Paresthesia and Movement Strategy

Overall, participants updated their movement strategies to account for the sensory inputs that were available to them. The results of Experiment 1 indicate participants updated their movement strategy according to if vision of the target and moving limb was or was not present, but only in the no-paresthesia condition. In contrast, when paresthesia was induced, participants did not adjust their movement time or time after peak velocity movement strategies. The lack of changes to MT and taPV when paresthesia was induced suggests that participants were not able to make use of online corrections to their limb trajectories as they normally would. Once again, this highlights the multisensory nature of these online corrections in that the corrections require both visual and proprioceptive inputs [2].

Based on the findings of Experiment 1, the combination of induced paresthesia and no visual feedback led to a significant movement bias towards the body midline. The findings also provide evidence for the idea that the contribution of proprioceptive input for movement accuracy is indeed larger in the absence of visual feedback. Specifically, the bias was greater with paresthesia when vision was removed. With intact sensory processing (i.e., without paresthesia and with full vision), participants had the longest MTs. In conjunction with improved endpoint accuracy, this pattern of results indicates that participants used the available sensory information and spent more time implementing online corrections to the limb trajectory. Notably, vision in the presence of induced paresthesia did not lead to longer MTs, presumably because the participants had less sensory feedback available to use for online movement corrections [2]. The longest MT in the intact sensory input condition was associated with the shortest time to peak velocity, especially with more practice. In the condition without vision and with paresthesia, the longer movement time (Figure 2b), accompanied by a marginal reduction in movement trajectory variability after 40% of the movement (which corresponds to approximately PV, and the onset of the limb-target correction phase; Figure 4a), as compared to the condition without vision and without paresthesia, could be explained by the principles of the optimized sub-movement model [42,45,46]. That is, due to the absence of availability of the two major sources of sensory input (visual and proprioceptive feedback), participants had to exchange a fast and forceful movement, which would lead to larger movement trajectory variability, for a more controlled movement, resulting in overall slower movements with less variability in movement trajectory. This choice of movement strategy fits with a ‘play it safe’ strategy and could account for the observed slower movements with longer movement times. In summary, the temporal movement findings are all in agreement that with full sensory information available and more practice, participants developed a new movement strategy. The differences in the temporal parameters in paresthesia versus no paresthesia conditions with full vision available signifies the role of proprioception for informing both the movement strategy and efficacy, even when visual information of the limb, target, and environment is available. 

Experiment 2 added augmented auditory feedback upon target acquisition. When auditory feedback was present then participants updated how they planned their movements (as seen through changes in RT, with a lack of any changes taPV) [23,37]. Changes in the time needed to initiate the movement (RT) as well as to execute the initial movement impulse (ttPV) are evidence of more efficient movement planning. Consistent with previous literature [47,48], it appears that the auditory feedback allowed participants to complete each movement more efficiently, freeing up processing resources to plan the next movement more effectively and efficiently. Further evidence of the improved planning includes the reduced trial-to-trial VE when paresthesia was induced and vision of the movement environment was removed.

### 4.3. Paresthesia and Movement Accuracy

The preceding section elaborated on the evidence found in both Experiments 1 and 2 for the significant contribution of proprioceptive input to the movement planning and strategies adopted by participants of both experiments. However, a similar effect was not found for movement accuracy. Specifically, the presence of visual feedback could compensate for the sensory feedback deficiencies caused by induced paresthesia for limb-target regulation after about 60% of the total movement time. This finding is supported by the results of the outcomes of CE (especially in the secondary axis) and VE (in both primary and secondary axes), which showed that vision was the most important source of sensory feedback for endpoint accuracy (Figure 3 and Figure 6). Furthermore, movement trajectory variability findings (Figure 4 and Figure 7) also displayed less variability after 60% into the movement in the full vision conditions, regardless of paresthesia. These findings are consistent with the literature in signifying the role of visual feedback as the dominant and most reliable source of spatial information for movements such as goal-directed aiming tasks [2,3,5], especially when participants are aware that visual information will be available to them [2,31]. The results of the present experiments demonstrate the key contribution of vision of the moving limb. Specifically, participants had difficulty updating their motor control strategies when both somatosensory and visual inputs were disrupted. In the context of models of limb control, such as the two-component multiple process model, this finding demonstrates that a minimum amount of (unaltered) sensory input is required for participants to update their movement strategies using pre-existing internal models of their limb movements.

### 4.4. Auditory Feedback and Movement Strategy

To further explore the influence of auditory feedback on movement performance and strategy under conditions of disrupted visual and somatosensory feedback, we conducted a mixed-design ANOVA (2 groups (auditory vs. no auditory) × 2 paresthesia (paresthesia vs. no paresthesia) × 2 practice (early vs. late performance)). We focused on comparison of performance outcomes for participants of Experiment 1 and Experiment 2, exclusively comparing conditions in which visual feedback was obstructed (no-vision conditions).

Our analysis centered on specific performance metrics aimed at assessing the impact of auditory feedback on both movement accuracy (CE, VE) and strategy (ttPV/MT, MT). The only significant finding was for the VE in the primary axis. The ANOVA analysis revealed a significant three-way interaction for the factors of paresthesia, group, and practice (F (1,24) = 5.547; *p* = 0.027, ηp^2^ = 0.188; Figure 8). Interestingly, the post hoc analyses revealed that only for the group who received auditory feedback was there a significant effect of practice (comparison of early vs. late trials) with paresthesia. That is, with paresthesia, participants had less VE with more practice only when auditory feedback was provided, while the same practice effect was not found with paresthesia in the group without auditory feedback. Also, the group without auditory feedback exhibited a significant practice effect (characterized by decreased endpoint variability) when paresthesia was absent, which was not the case for the group that received auditory feedback. This absence of practice effect for the group who received auditory feedback could be attributed to a potential ceiling effect influenced by the presence of auditory feedback.

Taken together, the present experiments provide further evidence that humans can improve movement consistency with augmented auditory feedback, which is especially relevant when intrinsic feedback is most disrupted. Notably, the present experiment extends previous findings from clinical populations to an experimental model of induced paresthesia. That is, when all other aspects of movement control remain typical, the temporary disruptions of somatosensory feedback led to identifiable differences in movement performance and the movement strategies used to achieve that performance. Consistent with previous research with clinical populations, improvements in movement performance with the addition of auditory feedback were seen through changes in movement planning. Interestingly, the strategy of adapting to induced paresthesia through changes in movement planning was also seen in our previous research where vision of the target was removed, but vision of the limb was still available. In the future, it will be interesting to assess if additional differences are evident if participants have more time to adapt to the induced paresthesia. 

It should be noted that another way to interpret the combined analysis is that auditory feedback supplemented for practice through a reduction in endpoint variability (but no change in other movement parameters). It has been proposed that augmented auditory feedback can free up the central nervous system for other processing—namely, improvements in movement planning for the subsequent movement [23,48,49]. Practice is another means by which sensory processing becomes more efficient, and these changes were seen through in the present experiment through more efficient taPV and improved endpoint accuracy with practice.

## 5. Limitations and Future Directions

With respect to applying the current results to upper limb rehabilitation, one limitation of the current study is the lack of a transfer test. Future research should incorporate a transfer test involving both compatible and incompatible sensory feedback sources to assess the specificity of the movement strategies participants adopt. This would aid in evaluating whether participants develop a reliance on the feedback following a short practice duration [15]. For example, a transfer task could explore how individuals who have practiced with vision and without paresthesia would adapt to scenarios of visual occlusion and/or paresthesia. Another limitation of the current design is that participants were exposed to the induced paresthesia for a relatively short duration of time. In many clinical cases, paresthesia is experienced over much longer durations of weeks, months, or years. Long duration exposure will undoubtedly lead to updated internal representations of the limb. Future research will need to examine how longer exposures to induced paresthesia affect limb control. Nonetheless, the present results provide new insights into how humans adapt to changing sensory inputs by inducing paresthesia in an otherwise healthy nervous system. Understanding how humans adapt to varying sensory inputs will support the development of more efficient, evidence-based rehabilitation programs.

## 6. Conclusions

We report that temporarily induced paresthesia influenced visuomotor control in an otherwise neurotypical adult group of participants. This finding is consistent with humans using proprioception of their current limb position to update movement control based on current and expected sensory consequences. When no auditory feedback was presented, then changes were observed primarily during movement execution (i.e., MT and time after PV), suggesting that participants were trying to adjust movement control by adapting parameters of their movement execution. In contrast, when auditory feedback of target acquisition was presented in Experiment 2, then participants instead updated how they planned through movements through changes in RT and time to PV.

The present experiments provide additional evidence that humans are able to flexibly adapt their movement strategies according to the presence and quality of different sources of intrinsic and augmented sensory feedback. Specifically, how humans adapt their approach to movement control depends on the combination of intrinsic and augmented feedback available. We report that participants updated their approach to movement planning or execution in specific ways that allowed them to achieve the task goals according to the sensory conditions. For example, through changing movement plans and/or taking time to update the movement trajectory based on available somatosensory and/or visual feedback. In other words, how humans change their movement strategy depends on the available sources of sensory input. The somatosensory manipulation used in the present experiments is relevant to various clinical populations who experience sensations such as paresthesia in the upper limb, including individuals with conditions such as stroke, spinal cord injury, or diabetes. The findings of this study can contribute to our understanding of how humans compensate for disrupted sensory inputs as well as how augmented auditory feedback may help to compensate for these disruptions. 

## Figures and Tables

**Figure 1 brainsci-13-01302-f001:**
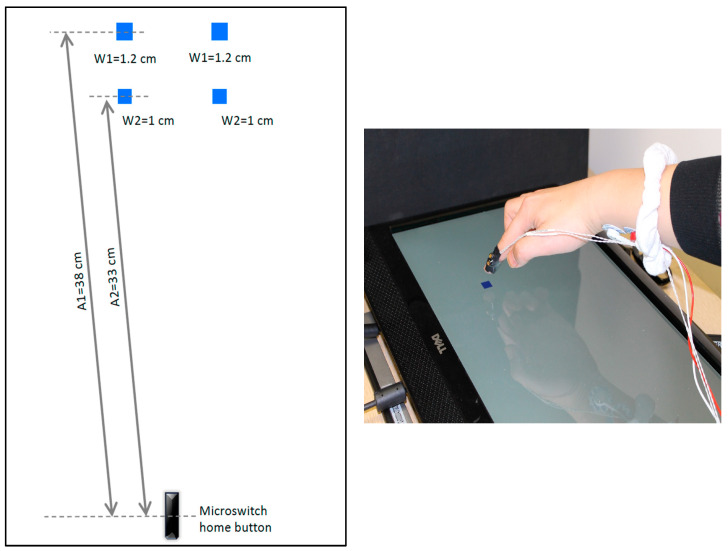
Goal-directed aiming task set-up (**right**) and 4 possible targets (**left**) with different amplitudes (A) and widths (W); all four possible targets had an index of difficulty of 6.

**Figure 2 brainsci-13-01302-f002:**
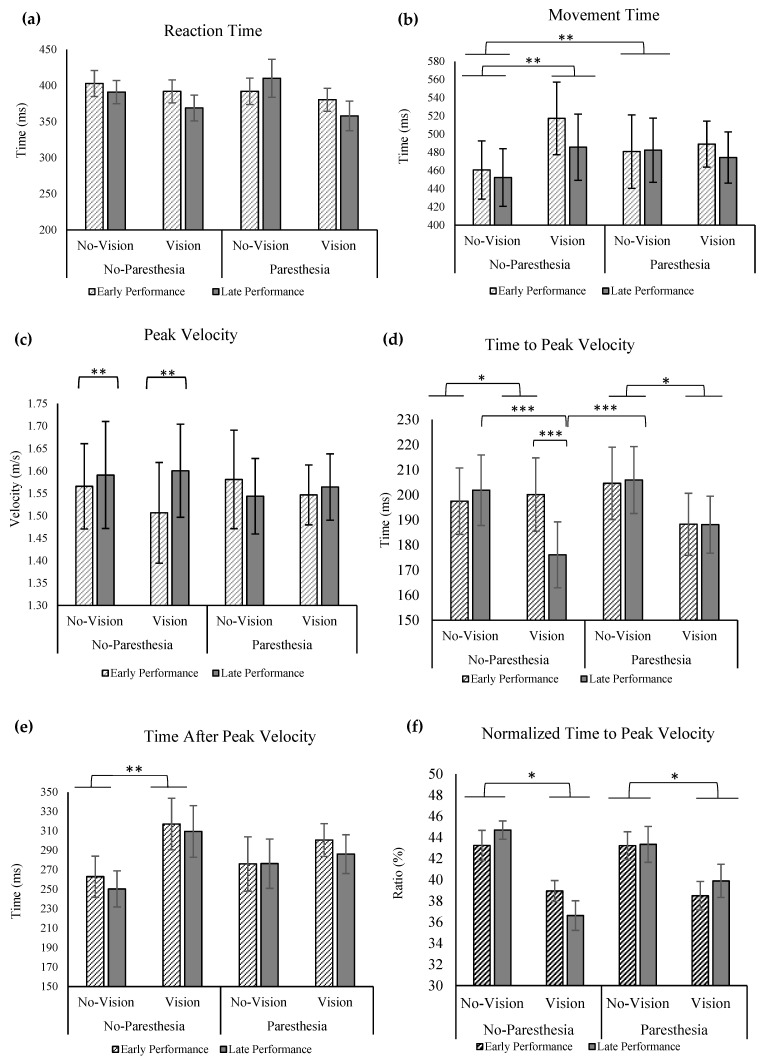
(**a**) Reaction time; (**b**) movement time with significant interaction of vision and paresthesia (**); (**c**) peak velocity with significant interaction of practice and paresthesia (**); (**d**) time to peak velocity with significant main effect of vision (*), interaction of vision and practice, and interaction of vision, practice, paresthesia (***); (**e**) time after peak velocity with significant interaction of vision and paresthesia (**); (**f**) normalized time to peak velocity (ttPV/MT) with significant main effect of vision (*). All error bars indicate standard error.

**Figure 3 brainsci-13-01302-f003:**
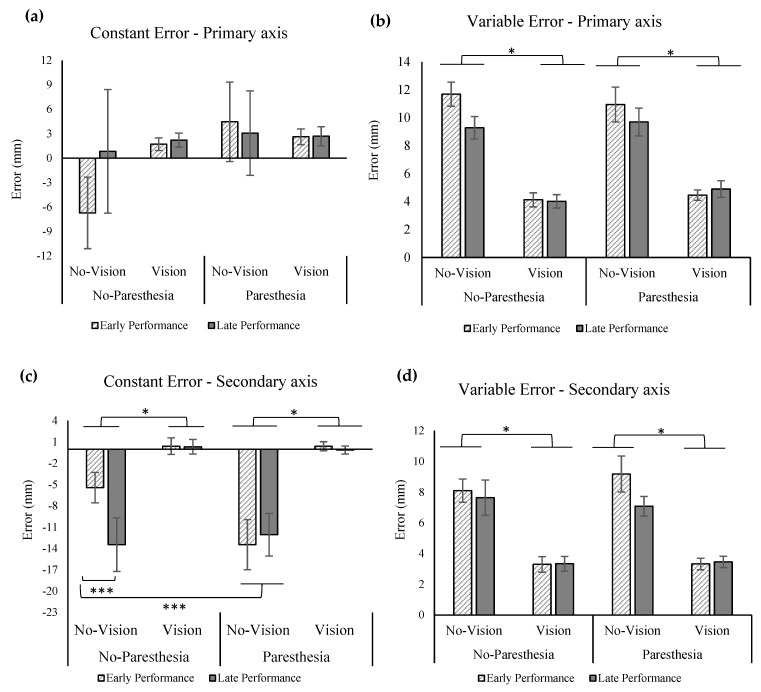
(**a**) Constant error in the primary axis (anterior–posterior); (**b**) variable error in the primary axis with significant main effect for vision (*); (**c**) constant error in the secondary (medio-lateral) axis with significant main effect of vision (*), interaction of paresthesia and practice, and interaction of paresthesia, practice, and vision (***); (**d**) variable error with significant main effect of vision (*). Negative values for constant error indicate undershoot errors. All error bars indicate standard error.

**Figure 4 brainsci-13-01302-f004:**
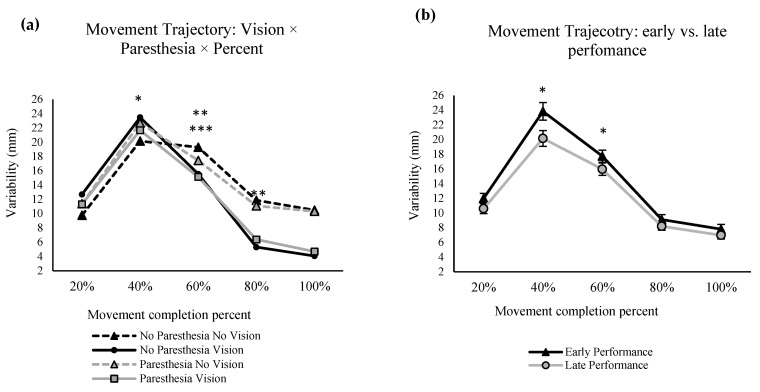
Spatial variability at different percentage of movement time in the primary axis of movement time. Spatial variability with significant main effects of vision and percent of movement (*, (**a**)) and interaction of vision and percent of movement (**, (**a**)); significant main effect for the factor of practice (*, (**b**)), two-way interaction of practice and percent of movement (**, (**b**)), and three-way interaction of vision, paresthesia, and percent of movement (***, (**a**)). The error bars are standard error.

**Figure 5 brainsci-13-01302-f005:**
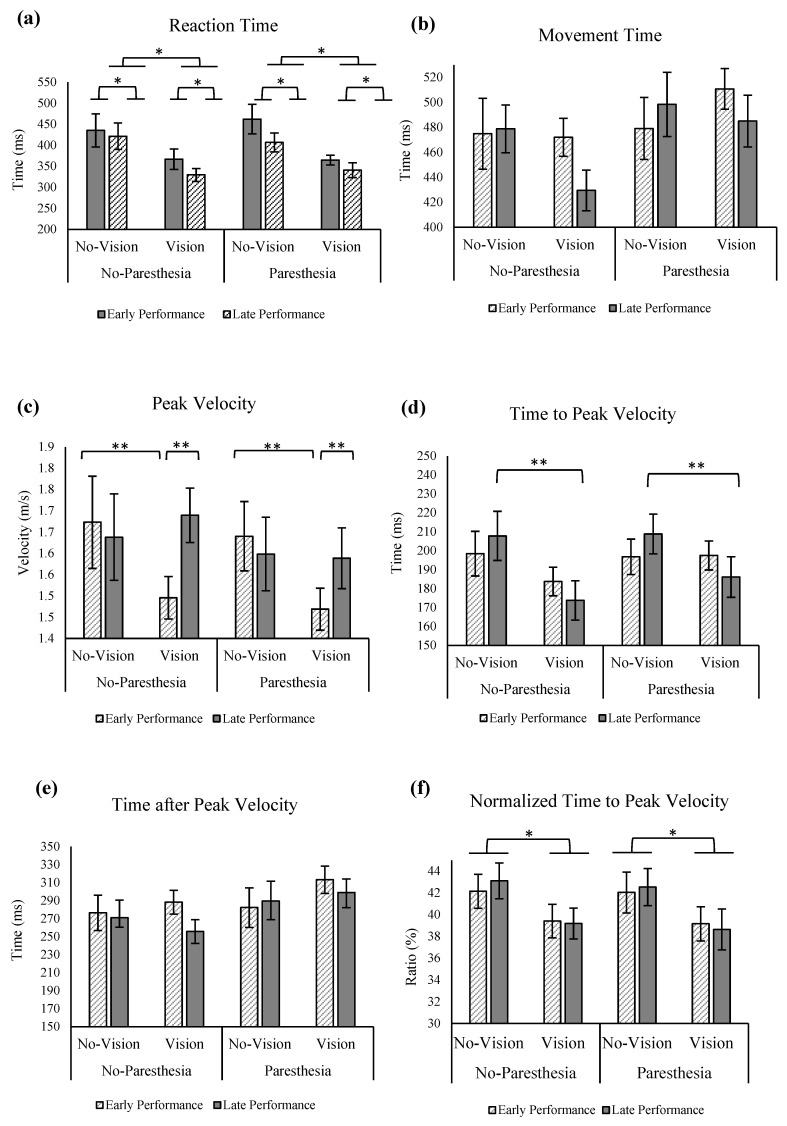
(**a**) Reaction time with significant main effect of vision and practice (*); (**b**) movement time; (**c**) peak velocity with significant interaction of practice and vision (**); (**d**) time to peak velocity with significant main effect of vision, and interaction of vision and practice (**); (**e**) time after peak velocity; (**f**) normalized time to peak velocity (ttPV) with significant main effect of vision (*). All error bars indicate standard error.

**Figure 6 brainsci-13-01302-f006:**
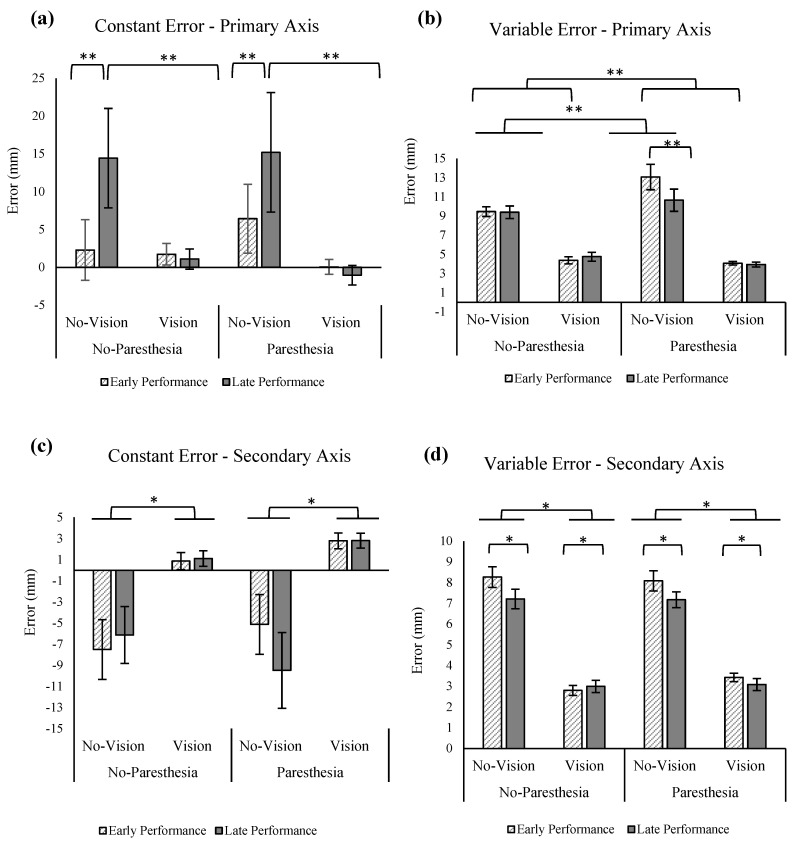
(**a**) Constant error in the primary axis (anterior–posterior) with significant main effect for vision and interaction between vision and practice (**); (**b**) variable error in the primary axis with significant main effect for vision, interaction of vision, and paresthesia, and interaction of vision and practice (**); (**c**) constant error in the secondary axis (medio-lateral) with significant main effect of vision (*); (**d**) variable error with significant main effects for practice and vision (*). Negative values for constant error indicate undershoot errors. All error bars indicate standard error.

**Figure 7 brainsci-13-01302-f007:**
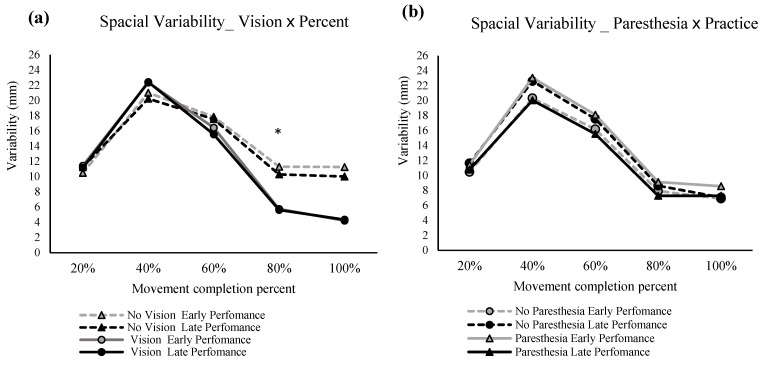
Spatial variability at different percentages of movement time in the primary axis of movement time. Spatial variability with significant main effects of vision and percent of movement (**a**) and interaction of vision and percentage of movement (*, (**a**)); significant interaction of practice and paresthesia (**b**).

**Figure 8 brainsci-13-01302-f008:**
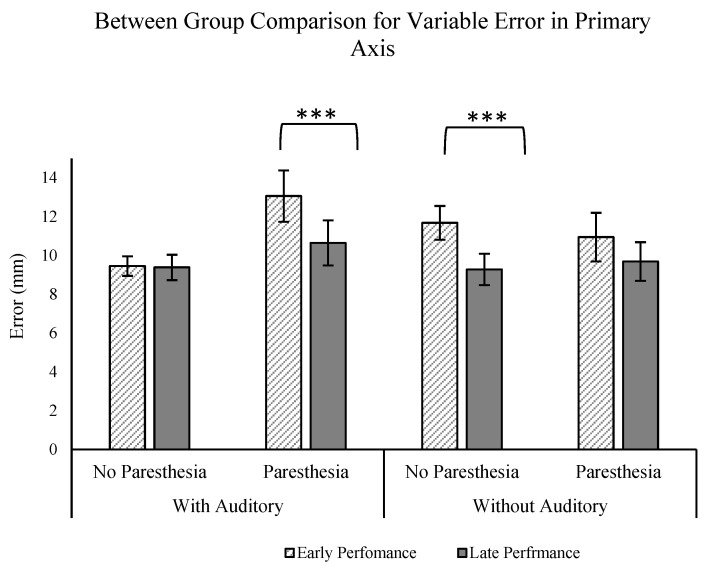
Between-group comparison of the auditory and no-auditory feedback groups for the outcome of VE in the primary axis: significant three-way interaction of factors of group, paresthesia, and practice was found (***). All error bars indicate standard error.

**Table 1 brainsci-13-01302-t001:** Monofilament pressure sensitivity test results for Experiment 1; sensed monofilament (grams) before induced paresthesia and after induced paresthesia right before the condition with paresthesia.

Participant	Sensed Monofilament Number	
	without Paresthesia	with Paresthesia
P01	2.83	3.61
P02	2.83	3.61
P03	2.83	3.61
P04	2.83	3.61
P05	2.83	3.61
P06	2.83	3.61
P07	2.83	3.61
P08	2.83	3.61
P09	2.83	3.61
P10	2.83	3.61
P11	3.61	4.31
Median	2.83	3.61

**Table 2 brainsci-13-01302-t002:** Monofilament pressure sensitivity test results for Experiment 2; sensed monofilament (grams) before induced paresthesia and after induced paresthesia right before the condition with paresthesia.

Participant	Sensed Monofilament Number	
	without Paresthesia	with Paresthesia
P01	2.83	4.31
P02	2.83	3.61
P03	2.83	3.61
P04	2.83	6.65
P05	2.83	3.61
P06	2.83	3.61
P07	3.61	4.56
P08	2.83	3.61
P09	2.83	4.31
P10	2.83	4.31
P11	3.61	4.56
P12	2.83	3.61
P13	3.61	6.65
P14	3.61	4.31
Median	2.83	4.31

## Data Availability

The data presented in this study are available on request from the corresponding author.

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
