# Peer review of "Optimizing Movement Performance with Altered Sensation: An Examination of Multisensory Inputs"

_brainsci, 2023, doi:10.3390/brainsci13091302_

Round 1

Reviewer 1 Report

Reviewer Comments

General comments:

The current study builds upon previous work surrounding the independent and integrative effects of visual and proprioceptive feedback. Here, within a rapid manual aiming task, visual and proprioceptive feedback are systematically manipulated using occlusion goggles and transcutaneous current stimulation, respectively. Generally speaking, the findings reveal an endpoint aiming bias following the removal of vision, while participants develop a strategy to more greatly use visual feedback whenever it becomes available (Exp. 1). The inclusion of augmented feedback appeared to enhance the endpoint precision, while leading to an opposing set of endpoint biases (Exp. 2).

The methods, data handling and analysis are robust, and the manuscript is well-written overall. However, there are some key points to consider including the emphasis on the importance or relevance of the present study, as well as further elaborating on the available data and subsequent explanations that are attributed to it. In addition, I have highlighted some other minor comments that mostly pertain to points of clarification.

Specific comments:

Introduction:

Current study is leveraged as a natural progression from a previous study that showed vision of the target and proprioception contribute toward endpoint accuracy. That is, it uniquely removes vision of the entire environment including the limb trajectory. While it would perhaps be ideal for the present study to systematically decouple visual feedback of the limb and target, it is possible to overlook this pitfall by providing at least some indication of how visual feedback of the limb is relevant.

For example, the role of peripheral vs. central vision (or visual asymmetries given the stereotypical oculomotor patterns in manual aiming) (e.g., early work of Bard/Paillard (pre-2000), and more recent central/peripheral visual manipulations from Khan and Proteau (post-2000)), and potential control differences with respect to the multiple processes model that was previously mentioned (Elliott et al., 2017). Along these lines, are there any known differences in aiming when vision of the limb is present or absent? By explaining this context, it will hopefully provide more of a foundation for the novelty and importance of the present study.

Experiment 1:

Discussion:

Pg. 10, Ln. 301-302 – It states that more time was devoted for conditions without vision compared to vision, although this is simply not true and in fact the opposite (!).

Pg. 11, Ln. 320-328 – Unless I’ve missed something, it’s suggested that the longer time after peak velocity when there is visual feedback can become less when there is paresthesia introduced, although this is not apparent within the report, nor is it observable within Figure 3.

It could be worth a mention of how the longer movement time for no vision with paresthesia compared to no paresthesia (Figure 3b) may be attributed to an attempt to sustain a reasonable degree of variability for the former as supported by the marginally smaller trajectory variability (Figure 4a). N.B., This fits with the suggestion surrounding optimization for enhanced aiming outcomes (Meyer et al., 1988; Hamilton & Wolpert, 2002; Elliott et al., 2004).

Experiment 2:

I may have missed it, but given the stated importance of terminal augmented feedback, I assume there was none present within Experiment 1 (i.e., occlusion goggles were opaque). Please clarify to what extent individuals were able to see the environment, and particularly the end-point limb response.

Results:

We could do with knowing the number of times individuals received the tone (synonymous with hitting the target). With this in mind, there could be a confounding influence in the frequency of feedback (e.g., more feedback received for vision compared to no vision).

Discussion:

Pg. 18, Para. 1 – Provide some supporting evidence and citations to support the claims being made about visual feedback provision and aiming strategy.

General Discussion:

The fact that vision helped overcome any paresthesia it seems when reviewing CE (specifically the secondary axis) and VE findings (Figure 3 and 6), it may highlight the importance and near dominance of this particular source of sensory feedback. Likewise, it appears vision is driving the differences in the trajectory variability findings (Figure 4 and 7). Further discussion including supporting theory and/or previous evidence is warranted.

Because of the element of practice, where there is some adaptation (i.e., early-late performance differences) when vision is present without induced paresthesia, it could be attributed either to the enhanced use of these sensory feedback sources during movement, or individuals having grown accustomed to the task owing to the offline benefits served by this feedback over a number of trials (Khan & Franks, 2003). The inclusion of a transfer test with both compatible and incompatible sensory feedback sources would help in this regard (e.g., Specificity of Practice paradigm; Proteau et al., 1987, 1992). For example, how might individuals who practice with vision and no paresthesia respond all of sudden to occlusion and/or paresthesia? Recognising this limitation including a relevant or more detailed discussion is warranted.

Minor comments:

Pg. 3, Para. 1 – There needs to be a clearer explanation of the manual aiming task itself. Specifically, there should be some indication of the different stages within an individual trial (e.g., preparation, potential ready signal, foreperiod, target presentation).

Assuming occlusion at movement onset, then there must have been some synchronised trigger signal like a press-and-release of a microswitch. If so, then this too should be highlighted.

Pg. 3, Ln. 107-108 – I may have missed it, but it states that data collection took place over two separate days, which I read as x2 conditions being completed on day 1 and 2 with each consisting of 200 trials (i.e., 100 trials per condition = 400 trial total). If so, this much should be made clearer to the reader.

Along these lines, it would help if the statement “Participants completed the four conditions in two separate experimental sessions” read as “Participants completed the four conditions across two separate experimental session”.

Pg. 3, Ln. 126 – Mention of the filter parameters for smoothing is missing some keys points including phase-lag (e.g., dual-pass) and order (e.g., 2nd-order).

Pg. 3, Ln. 135 – Spatial variability is taken as a function of the percentage of movement time. For an amplitude-related task, it is standard to extract this measure as a function of kinematic landmark (e.g., peak acceleration, peak velocity, peak negative acceleration, end), whilst a direction-related task (i.e., target passing/striking) customarily does it as a function of the entire limb displacement (e.g., 20%, 40%, etc of movement amplitude). What is the rationale or recourse for the current choice of percentage of movement time?

Pg. 3, Para. 4 – There should be mention of the intended effect size measure.

Figure 1 – The lines indicating amplitude would imply that the target amplitude for each of these targets was taken solely with respect to the anteroposterior plane (perhaps within the y-axis). This is somewhat misleading as there is fair contribution from the mediolateral plane as well (perhaps within the x-axis). Therefore, amplitude should be taken as the resultant or radial extent based on a combination of both planes (i.e., x- and y-axes). Please amend the nominal ID within the Methods as a result.

Pg. 5, Ln. 172-174, and Pg. 204-207– Because the two-way interaction is superseded by the three-way interaction, there is no real need to undertake and report on a post hoc for the former (i.e., only the latter three-way interaction is relevant).

Pg. 7 and 16 – Report on the CE in the primary (mediolateral) axis describes undershoot and overshoot effects, although it can become confusing when also referring to these terms in the context of CE in the secondary (anteroposterior) axis. Thus, alter the terminology so a more distinct separation can be formed between the different directions.

Along these lines, it should also be made clear that the primary and secondary axes each allude to the mediolateral and anterolateral planes, respectively. However, admittedly, I’ve probably missed this somewhere in the Methods.

Figures 2, 3, 5 and 6 – It could be an artefact of the manuscript draft production process, but the lines/brackets/bars hovering over the charts to indicate pairwise differences are somewhat askew, and therefore difficult to follow or interpret.

Figures 3 and 6 – I may have missed it in the Methods, but state the relation between signed error (+/-) and movement direction (medial/lateral) within the respective figure captions.

Pg. 18, Ln. 520-521 – More precisely, PV is reduced to limit or dampen the variability that could effect the endpoint accuracy (see also, earlier comments on optimization; e.g., Meyer et al., 1988).

Pg. 19, Para. 2 – In light of the absence of a more definitive explanation for the unexpected effect of no vision in later practice for CE of the primary (mediolateral) axis, it could be useful to observe the trial-by-trial adaptation (N.B., it’s possible to make a more coarse, but efficient observation of this by reviewing any major difference in the mean and median of CE values – if there is, then the profile is somewhat skewed and there may be some drastic adaptation going on). That is, it is possible that the originally stated overcompensation of more rightward and leftward responses could be drastically negated following the introduction of augmented feedback.

Likewise, any claims of an adaptive response following the augmented feedback may be examined by reviewing trials that followed on from those without (error) compared to with (no error) the feedback (e.g., Elliott et al., 2004).

References:

Elliott, D., Hansen, S., Mendoza, J., & Tremblay, L. (2004). Learning to optimize speed, accuracy, and energy expenditure: A framework for understanding speed-accuracy relations in goal-directed aiming. J Mot Behav, 36, 339-351. doi: 10.3200/JMBR.36.3.339-351

Hamilton, A. F., & Wolpert, D. M. (2002). Controlling the statistics of action: obstacle avoidance. J Neurophys, 87, 2434-2440.doi: 10.1152/jn.2002.87.5.2434

Khan, M. A., & Franks, I. M. (2003). Online versus offline processing of visual feedback in the production of component submovements. J Mot Behav, 35, 285-295. doi: 10.1080/00222890309602141

Meyer, D. E., Abrams, R. A., Kornblum, S., Wright, C. E., & Smith, J. E. K. (1988). Optimality in human motor performance: Ideal control of rapid aimed movements. Psychol Rev, 95, 340-370. doi: 10.1037/0033-295X.95.3.340

Reviewer 2 Report

This study examined how to optimize a movement performance with altered sensation using multisensory inputs. This study is quite intriguing, with good methodology and interesting findings. However, the manuscript writing needs quite a few improvements. Please find the comments and raised issues below.

Introduction:

1.     Overall, the introduction was written more like a book chapter rather than a scientific paper. For example, paragraphs 2 and 3 consisted of only four references. Authors focus more on elaborating on some basis rather than providing a literature review on this topic.

2.     Furthermore, a lot of claims are missing a citation. For example, “We found participants changed their movement strategy from online control to pre-planned movements when paresthesia was present, and vision of the target was removed.” I see that it relies on your previous work. However, you cannot write an entire paragraph about one previous paper.

3.     More than half of the Introduction is actually elaborating on experiments, while no rationale for this study was clearly presented, rather than: this is the continuation of our work. I agree, but this is a completely new paper, and you need a well-written Introduction and strong rationale to back up this (quite good) work.

4.     Finally, try to be more concise with the aims. Please add an aim, hypothesis, and contribution to the field.

Methods:

The methodology was extensively elaborated and well explained. Some minor issues can be addressed.

1.     Please divide section 2.1.2. into at least two section for better readability. This includes a distinction between data analysis and statistical analysis

2.     Please indicate the statistical software that was used for analyses.

Results:

1. Please avoid repeating post hoc analysis in the text if they are quite nicely presented in the graph. You can just write in one sentence they are in the graph. This will improve the readability of the results.

Discussion:

1. The discussion is very long and hard to read and follow, with a lot of repeating of the results. Also, there are too many comparing with previous studies, with less elaboration or interpretation of the results.

Conclusions:

1. Please avoid citation in the conclusions, referring to the previous findings. This should be your conclusions, based on the discussion.

2. Also, provide some practical applications and suggestions for future research.

Reviewer 3 Report

The study is very interesting and well structured. The purpose of the study is clear and the execution of the experiments is reported in detail. Furthermore, the statistical analysis used is appropriate. I have just a few minor suggestions for authors.

If possible the authors should try to simplify the figures trying to make them easily understandable. I also recommend moving the letters (a, b, c....) to another position as the reference is not clear. For example, they could be placed in the upper left corner.

I have another suggestion for authors which concerns paragraph 2.3 (Discussion). This paragraph begins: The aim of..... I think it would be more appropriate to start this section by talking about the results of the first experiment and then proceed to discuss them.

Round 2

Reviewer 2 Report

Dear authors,

I am pleased with your answers and corrections regarding my concerns and raised issues.

Best regards